# Microstructure and Corrosion Resistance of Underwater Laser Cladded Duplex Stainless Steel Coating after Underwater Laser Remelting Processing

**DOI:** 10.3390/ma14174965

**Published:** 2021-08-31

**Authors:** Congwei Li, Jialei Zhu, Zhihai Cai, Le Mei, Xiangdong Jiao, Xian Du, Kai Wang

**Affiliations:** 1College of Mechanical Engineering, Beijing Institute of Petrochemical Technology, Beijing 102617, China; m17854117942@163.com (C.L.); zhujialei@bipt.edu.cn (J.Z.); jiaoxiangdong@bipt.edu.cn (X.J.); 18810660456@163.com (K.W.); 2National Engineering Research Center for Remanufacturing, Army Academy of Armored Forces, Beijing 100072, China; zlbdy@163.com; 3Shanghai Nuclear Engineering Research and Design Institute, Shanghai 200233, China; meile@snerdi.com.cn

**Keywords:** underwater additive manufacturing, underwater laser remelting, phase transformation, corrosion resistance

## Abstract

Combined with the technologies of underwater local dry laser cladding (ULDLC) and underwater local dry laser remelting (ULDLR), a duplex stainless steel (DSS) coating has been made in an underwater environment. The phase composition, microstructure, chemical components and electrochemical corrosion resistance was studied. The results show that after underwater laser remelting, the phase composition of DSS coating remains unchanged and the phase transformation from Widmanstätten austenite + intragranular austenite + (211) ferrite to (110) ferrite occurred. The ULDLR process can improve the corrosion resistance of the underwater local dry laser cladded coating. The corrosion resistance of remelted coating at 3 kW is the best, the corrosion resistance of remelted coating at 1kW and 5kW is similar and the corrosion resistance of (110) ferrite phase is better than grain boundary austenite phase. The ULDLC + ULDLR process can meet the requirements of efficient underwater maintenance, forming quality control and corrosion resistance. It can also be used to repair the surface of S32101 duplex stainless steel in underwater environment.

## 1. Introduction

During the long-term service of nuclear power plants, the duplex stainless steel (DSS) plate of the spent fuel pool will have an aging effect. Its failure mechanism is mainly uniform corrosion, stress corrosion cracking (SCC) and pitting corrosion, which poses a serious threat to the safe operation of nuclear power plants [1]. Since the spent fuel pool generally works in the water environment, in order to reduce the equipment maintenance cost and consider the reasons of nuclear radiation, its repair is usually repaired by underwater welding technology. Underwater welding methods include underwater wet welding, local dry underwater welding and high-pressure dry underwater welding. After years of development, local dry underwater welding technology has been proved to be a better underwater repair technology [2,3].

Underwater laser cladding technology has the characteristics of low heat input and high repair accuracy, so it has gradually become an important key technology for the underwater repairs of nuclear power equipment. In the past decade, researchers have focused on underwater wet laser cladding (UWLC) and local dry underwater laser cladding (LDULC). The UWLC directly acts on the substrate in the underwater environment through the laser beam to prepare the coating by prefabricated powder or synchronous powder feeding. In the research of UWLC, Guo et al. [4] studied the influence of water depth on laser welding forming, and analyzed the formation mechanism of a “beam channel” when the laser beam acts on the substrate through water. Feng Xiangru et al. [5,6] found that the interactions of the laser, water and maintenance object affect the repair quality during the UWLC process. The homemade protective covering and the useful materials (as Ti) were used to solve the problems of UWLC. LDULC is used to drain the water on the substrate surface through the drainage nozzle to form a local dry cavity to prepare the coating. Guo et al. [7] designed a double-layer gas drainage nozzle for local dry underwater laser welding, and obtained a welded joint with low porosity. Fu et al. [8,9,10] successfully prepared 304 stainless steel coatings and welded joint and thin-walled Ti-6Al-4V parts by LDULC utilizing a drainage nozzle. According to previous studies, UWLC has poor forming, great influence by the water environment and low adaptability. LDULC can produce excellent coatings, and is therefore a good technical method for underwater environment repairs.

The microstructure of DSS is composed of ferrite (δ) and austenite (γ). Compared with austenitic stainless steel, DSS has excellent intergranular corrosion resistance [11]. In the study of heat treatment and laser surface treatment of DSS, Mohammad M et al. [12] found that the retained austenite in the grain of duplex stainless steel transformed into martensite during cold rolling. Zhang et al. [13] studied the temperature evolution, comparison and corrosion resistance of DSS treated by a laser surface heat treatment. Zhu et al. [14] investigated the occurrence of corrosion during the sub rapid solidification of DSS δ→δ + γ phase transformation. Zhang et al. [15] studied the transformation from face-centered cubic austenite (fcc) to body-centered cubic (bcc) ferrite of duplex stainless steel during in-situ solid solution annealing heat treatment. Alex M. et al. [16] studied the laser surface melting treatment of DSS, which resulted in increased surface hardness and decreased corrosion resistance. M.B.mampuya et al. [17] studied the effect of the water-cooled DSS microstructure after solution annealing at 1100 °C and found that austenite invaded ferrite. Lin et al. [18] studied water cooling after laser surface remelting and found that grain refinement, precipitation strengthening, dislocation strengthening and a mixed microstructure occurred.

Tae Woo H. et al. [19] studied an underwater wet laser remelting process to better control the surface quality of selective laser melting (SLM) parts. Xin Bo et al. [20] used mixed laser remelting and laser metal deposition process to form thin-walled structure with 316L steel powder. It was found that the hybrid process can significantly improve the ultimate tensile strength and yield strength of the composite. Stefan K. et al. [21] studied that the combination of laser cladding and laser remelting can be used to prepare single-crystal structures.

As mentioned, ULDLC is an excellent technical means for solving the problem of underwater environment repairs. Laser remelting can be used not only to improve the metal surface structure, but also to repair metal surface defects. However, in the aspect of experimental equipment setting, the existing research has been limited to investigations using water depths of less than 50 mm. The shallow water cannot simulate the real underwater repair environment and the cooling gradient of the molten pool. In terms of technology, although the excellent forming coating was prepared, its performance was not improved because of the particularity of the underwater environment. In this paper, a waterproof laser cladding head was developed. Combined with technology of ULDLC and ULDLR, the laser cladding and laser remelting experiments of S32101 DSS, a plate material of spent fuel pool, were carried out in underwater environment. The microstructure and electrochemical corrosion properties of the coating were studied.

## 2. Process Experiment

The S32101 duplex stainless steel was used as the basic material (BM) with the original size of 300 mm × 150 mm × 16 mm. The chemical compositions of the S32101 DSS used in this research were displayed in Table 1 from the manufacturer (Ansteel, Anshan, China). The filler material was ER-2209 wire (Kunshan Gintune Welding, Suzhou, China). with 1.2 mm diameter, which was delivered by wire feeder and passed through the underwater local dry laser cladding nozzle (ULDLCN), and the chemical compositions of ER-2209 filler wire were displayed in Table 2 from the manufacturer. Before the experiment of ULDLC, the substrate was roughened with a steel wire brush in order to decrease its reflectivity to laser radiation and then cleaned using alcohol and acetone in an ultrasonic cleaner to remove the surface contaminants. ULDLC/ULDLR was carried out by using an underwater laser cladding system shown in Figure 1 and Figure 2, consisting of an RFL-6000 laser (Wuhan Raycus, Wuhan, China), a waterproof laser cladding head, an ULDLCN and a computer controlled three-axis positioning system. During experiments ULDLC and ULDLR, the DSS plate was placed 500 mm underwater, and the water temperature in the water tank was 25 °C. After many orthogonal experiments in the early stage, the parameters of laser cladding were as follows: laser power—5 kW, laser spot diameter—5 mm, laser moving speed—10 mm/s, gas flow rates of ULCN—50 L/min. Parallel laser tracks with 48% overlap were fabricated to form a coating over the whole BM and the obtained sample was denoted an as-cladded DSS coating. After laser cladding, the laser remelting was also processed using the laser cladding system with parameters as follows: laser power—1 kW, 3 kW and 5 kW, laser spot diameter (3 mm) and a laser moving speed of 10 mm/s. Thus, these obtained samples were named as as-remelted-1 kW DSS coating, as-remelted-3 kW DSS coating and as-remelted-5 kW DSS coating, respectively. Figure 3 illustrated the underwater laser cladding route, and the underwater laser remelting process diagram.

After ULDLC and ULDLR, the metallographic specimens were cut along the Y axis from the DSS thick-walled part specimens in the two environments, and then the cold mounting method was used for inlaying. After polishing with SiC sandpaper and polishing with abrasive spray, the sample was subjected to metallographic etching treatment with FeCl_3_-HCl solution. The phase compositions of the as-cladded DSS coating and as-remelted DSS coatings were analyzed by the X-ray diffraction technique (XRD, Bruker Advance D8, Bruker, Karlsruhe, Germany). Microstructure observations were performed on Optical microscope (OM, Leica, Wezlar, Germany). The elements distribution characteristics in austenite and ferrite phases within the selected areas of the etched samples were studied by an electron probe micro-analysis (EPMA-1600, Shimadzu, Kyoto, Japan) equipment. Electrochemical performances of cladding coatings in 3.5 wt.% NaCl solution were investigated by a three-electrode workstation (VersaSTAT 3F, Ametek, Oak Ridge, America), consisting of a working electrode referring to the tested sample of the cladding layer, a Saturated Calomel Electrode (SCE) as the reference electrode and a platinum electrode as the auxiliary electrode. Potentiodynamic polarization curves were measured with the scanning rate of 0.2 mV/s, while the scanning potential ranged from −0.8 to +1 V. An electrochemical impedance spectroscopy (EIS) plot was obtained within the frequency from 10 mHz to 10 kHz whilst the amplitude was 10 mV. The localized electrochemical measurements were performed through a Versa SCAN scanning electrochemical workstation (Ametek, Oak-Ridge, America), scanning electrode vibration technology (SEVT) module was used to measure the corrosion potential and corrosion resistance of the coatings in 3.5 wt.% NaCl solution.

## 3. Results and Discussion

### 3.1. Coating Appearance and Characteristics

Figure 4 shows the macro morphology and cross-section of the as-cladded coating and as-remelted coating at different laser power values (1, 3 and 5 kW). The length of coatings was 100 mm. During the underwater laser cladding process, the solidification of the molten pool had a faster cooling rate and no oxidation. Therefore, the appearance of all coatings were continuous and uniform, with no obvious defects such as cracks, pores, inclusions, or lack of fusion, and the color was silver-white. Compared to the as-cladded coating, the underwater laser remelting process eliminated the fish-scale phenomenon on the surface of the coating. As the laser power density increased, the surface metal gloss of the as-remelted coating was better and the surface roughness was smaller. By analyzing the cross-section of the DSS coating, it was found that as the laser power increased, the depth of the laser remelted affected zone gradually increased, as displayed in Table 3. When the laser remelting parameter was 5 kW, and the penetration depth of the remelting affected zone (RAZ) exceeded that of the original coating. When the laser remelting parameter was greater than 3 kW, a large number of dendrites grew and were distributed along the direction perpendicular to the boundary of the molten pool, because this direction was the largest temperature gradient and thus the heat dissipation was the fastest.

### 3.2. Phase Compositions

Phase compositions of the as-cladded coating and as-remelted coatings were analyzed by XRD technique. The X-ray diffraction patterns presented in Figure 5 reveal that the as-cladded coating consists of γ and δ phases, with four γ peaks {(111), (200), (220) and (311)} and four δ peaks {(110), (200), (211), and (220)}, respectively. This result is consistent with other studies [18]. However, the (211) δ phase peak of the as-cladded coating is the highest, and the four γ peaks {(111), (200), (220) and (311)} are lower. This suggests that the formation of γ phase is inhibited under the condition of circulating water cooling, as the γ has less time to precipitate from the matrix δ. With the increase of laser power density, the peak of {(211)} δ phase decreases, the peak of {(110)} δ phase first decreases and then increases. The peaks of {(111), (220), and (311)} γ phase increase obviously in the as-remelted-1 kW DSS coating.

Figure 6 reveals that OM images of as-cladded and as-remelted DSS coatings. The interface of the coating/substrate is visible (see Figure 6c,f,i,l) and the fusion line can also be seen clearly, which indicates a good metallurgical bond of the coating with the substrate. The microstructure was analyzed according to reference [15]. The microstructure of the as-cladded DSS coating is composed of Widmanstätten austenite (WA), grain boundary austenite (GBA), intragranular austenite (IGA) and sheet ferrite, as is shown in Figure 6a,b. The remelting zone of as-remelted-1 kW DSS coating consists of coarse grained δ, sheet ferrite and a large number of secondary austenite (γ_2_), it corresponds to the test results of XRD, as is shown in Figure 6d. It shows that low laser power remelting is conducive to the formation of a secondary austenite phase. The microstructure of as-remelted-3 kW/as-remelted-5 kW DSS coatings consists of fine grain δ, coarse grain δ, GBA and a small amount of WA and IGA. The recrystallization of the coating is caused by underwater laser remelting. The contents of ferrite, WA and IGA in the underwater laser remelted zone decrease obviously compared with the non-underwater laser remelted zone, and the contents of single phase δ increase obviously. Underwater laser remelting causes the coating to recrystallize, and the δ with different grain size appears due to the uneven cooling rate in different zones. Fine grain δ appears at the top of the as-remelted coatings, as shown in Figure 7a, and coarse grain δ appears at the middle of the coating, as shown in Figure 7b. With the increase of remelting laser power density, the heat affected zone area of laser remelting increases (Figure 6c,f,i,l). δ (black part) appears in the heat affected zone due to the cyclic input of laser energy.

### 3.3. Distribution Characteristics of Alloying Elements

In order to determine the effect of underwater laser remelting on the elements distribution characteristics in the austenite and equiaxed ferrite phases, the sample have been examined by EPMA. It can be seen from Figure 8 that elements Cr mainly concentrate in the δ phase, while the peak value of Ni content tends to stabilize in γ phase, which is consistent with previous literature [22]. It is further confirmed that the equiaxed grains formed after underwater laser remelting are ferrite.

### 3.4. Mechanism of WA + IGA + (211) δ→(110) δ Phase Transformation

The laser remelting process is a local quenching and heating process. Owing to the high heating speed, the surface heating rate can reach 10^4^ °C–10^8^ °C/s. The surface of the material reaches the austenitizing temperature rapidly, and the ferrite is transformed into austenite through non-diffusion [18]. However, in the underwater wet laser remelting process, circulating water is used as the cooling medium, and the cooling rate of the molten pool is faster. After underwater laser remelting, the acicular martensitic α′ needles, that can be considered as secondary structures, were pronounced, which probably grew because of the subsequent heating and cooling in the water [19].

The microstructure and phase composition of the as-remelted coatings were discussed. The microstructure of remelted 1 kW coating evolves and a grain boundary appears. The heating temperature does not completely reach the grain homogenization temperature, which indicates that the new grains begin to grow. When the laser remelting power is more than 3 kW, the contents of WA, IGA and flaky ferrite in the top and middle of the cladding layer decrease obviously, and equiaxed grains appear. Equiaxed grains are surrounded by GBA grains with different grain sizes. The phase transformation of face-centered cubic austenite (fcc) to body-centered cubic (bcc) ferrite in duplex stainless steels during in-situ solution annealing heat treatment [15]. Combined with the phase composition and chemical element distribution of the remelted coating, it can be inferred that the equiaxed ferrite appears after underwater laser remelting.

Equiaxed ferrite consists of fine and coarse grains. This is due to the high cooling rate at the top and the relatively low cooling rate at the middle, which leads to the influence of different degree of under cooling during ferrite growth. Therefore, it is inferred that when the laser remelting power is greater than 3 kW, the WA, IGA and flaky ferrite are transformed into equiaxed ferrite. The whole ferrite transformation process: equiaxed grain growth, WA + IGA + (211) δ→(110) δ, (110) δ homogenization. The schematic diagram is shown in Figure 9.

### 3.5. Corrosion Performance

The electrochemical characterizations of as-cladded and as-remelted DSS coatings in 3.5 wt.% NaCl solution were analyzed and the corresponding potentiodynamic polarization curves and electrochemical impedance spectroscopy (EIS) data in the form of Nyquist plot were presented in Figure 10. Electrode potential (E) and polarization electric current (I) were tested. As displayed in Figure 10, the polarization curves of both the as-cladded and as-remelted DSS coatings displayed the same trend, but the as-cladded coating’s polarization curve located in the lower region. As displayed in Table 4, the as-cladded coating’s corrosion potential was −283.41 mV, lower than that of as-remelted coatings. It indicated that the as-cladded coating was easier to be corroded than as-remelted coatings. As displayed in Table 4, the order of self-corrosion potential: as-remelted-3 Kw > as-remelted-5 kW > as-remelted-1 kW > as-cladded. The self-corrosion current density of the as-cladded coating is lower than that of the as-remelted coatings, means that the as-cladded coating possessed a slightly higher corrosion rate compared to as-remelted coatings.

As displayed in Figure 11, Z′ and Z″ as the real and imaginary parts of the measured impedance Z. The impedance spectrum radius of as-cladded coating was smaller than that of the as-remelted coatings obviously, it is thus possible to infer that there is poor corrosion resistance of the coating not remelted by laser. The results show that the underwater laser remelting process can improve the corrosion resistance of the as-cladded coating. The as-remelted-3 kW coating has the best corrosion resistance, and the corrosion resistance of the as-remelted-1 kW and as-remelted-5 kW coating is similar.

As described in Section 3.2, in the surface area of the coating without laser remelting, the grains are composed of WA, GBA, IGA and flaky ferrite. After underwater laser remelting, recrystallization occurs in the same coating area, and the grains are composed of a small amount of WA, IGA and GBA equiaxed ferrite. The results show that the content of equiaxed ferrite increases and the content of WA and IGA decreases, which improves the corrosion resistance of the as-remelted coatings.

In order to further ascertain the corrosion resistance between GBA and (110) δ on the as-remelted-3 kW. SVET measurement was performed on the as-remelted-3 kW sample. Figure 12 shows the result of the SVET measurement and microstructure of the scanning area. It can be seen that there exists two electric potential (E) terraces far from the borderline between GBA and (110) δ phases. The higher E zone, which corresponds to GBA phase, is about 0.4–1.8 μV. while the lower E zone, which corresponds to (110) δ phase is just about −3.5 μV. The lower the E is, the better of the corrosion resistance is, this indicating that the corrosion resistance of (110) δ phase is better than that of GBA phase.

## 4. Conclusions

By using the developed underwater laser cladding system, underwater laser cladding and underwater laser remelting on laser cladding layers with different laser energy densities was carried out. The mechanism of WA + IGA + (211) δ→(110) δ phase transformation and its electrochemical corrosion performance has been investigated. The major conclusions are summarized in the following:

1. The underwater laser remelting process eliminates the fish scale phenomenon on the surface of the as-cladded coating. When the laser power density increases, the surface metal gloss of the as-remelted coating is better, the surface roughness is smaller and the depth of laser remelting influence zone increases gradually. When the laser remelting power is greater than 3 KW, a large number of dendrites grow and distribute along the direction perpendicular to the molten pool boundary.

2. The microstructure of the as-cladded coating is composed of WA, GBA, IGA and flaky ferrite. The recrystallization of the coating is caused by underwater laser remelting. The contents of (211) ferrite, WA and IGA decrease obviously and the contents of (110) δ increase obviously in laser remelting zone. During the recrystallization process, different regions with different degrees of undercooling form different (110) δ grain sizes.

3. When the laser remelting power is greater than 3 kW, the WA, IGA and flaky ferrite are transformed into equiaxed ferrite. The whole ferrite transformation process: equiaxed grain growth, WA + IGA + (211) δ→(110) δ, (110) δ homogenization.

4. The results show that the underwater laser remelting process can improve the corrosion resistance of the as-cladded coating. The as-remelted-3 kW coating has the best corrosion resistance, and the corrosion resistance of the as-remelted-1 kW and as-remelted-5 kW coating is similar. The (110) δ phase has more corrosion resistant than that of GBA phase.

## Figures and Tables

**Figure 1 materials-14-04965-f001:**
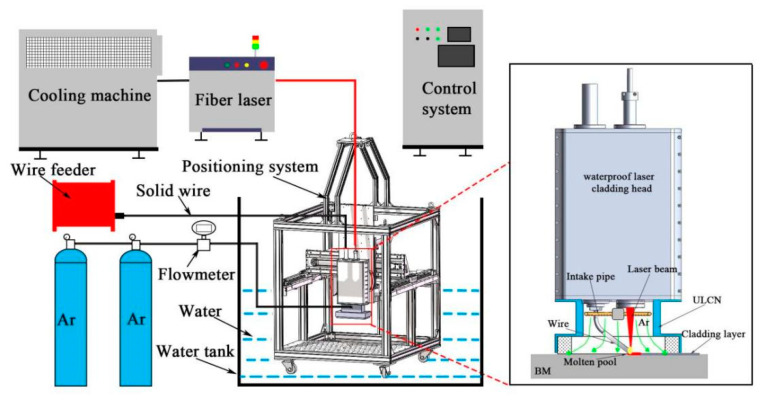
Schematic of underwater laser cladding system.

**Figure 2 materials-14-04965-f002:**
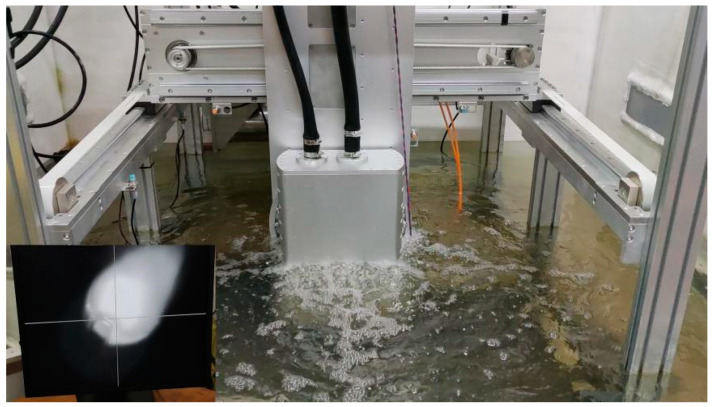
Underwater laser cladding system for underwater laser cladding/remelting processing.

**Figure 3 materials-14-04965-f003:**
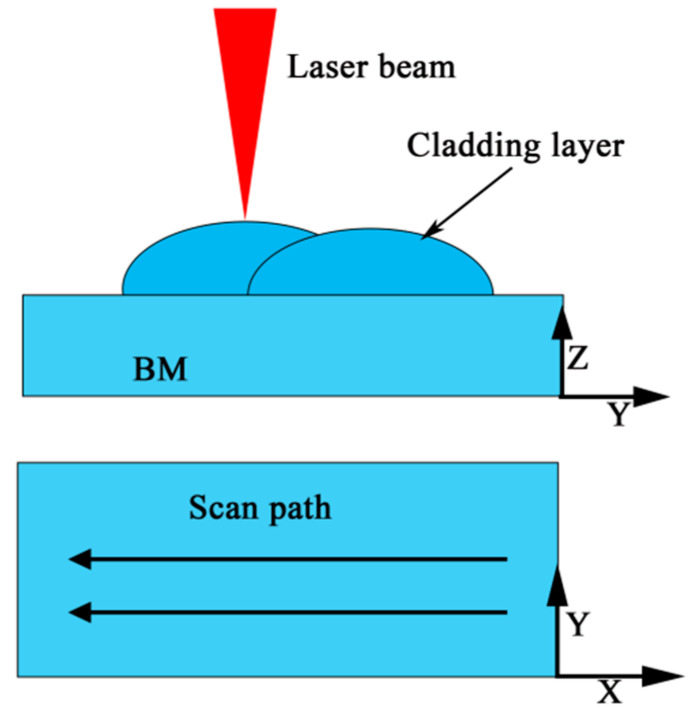
Diagram of the underwater laser cladding.

**Figure 4 materials-14-04965-f004:**
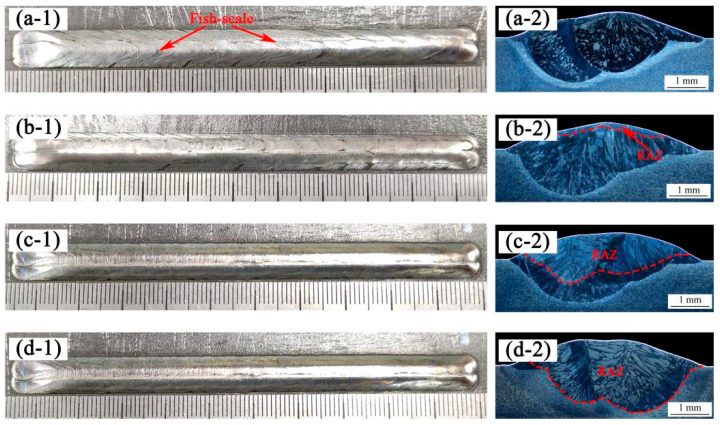
Macroscopic morphology of laser cladding coating: (**a****-1**) as-cladded DSS coating, (**b****-1**) as-remelted-1kW DSS coating, (**c-1**) as-remelted-3kW DSS coating, (**d-1**) as-remelted-5kW DSS coating. Cross-section of laser cladding coating: (**a****-2**) as-cladded DSS coating, (**b****-2**) as-remelted-1kW DSS coating, (**c-2**) as-remelted-3kW DSS coating, (**d-2**) as-remelted-5kW DSS coating.

**Figure 5 materials-14-04965-f005:**
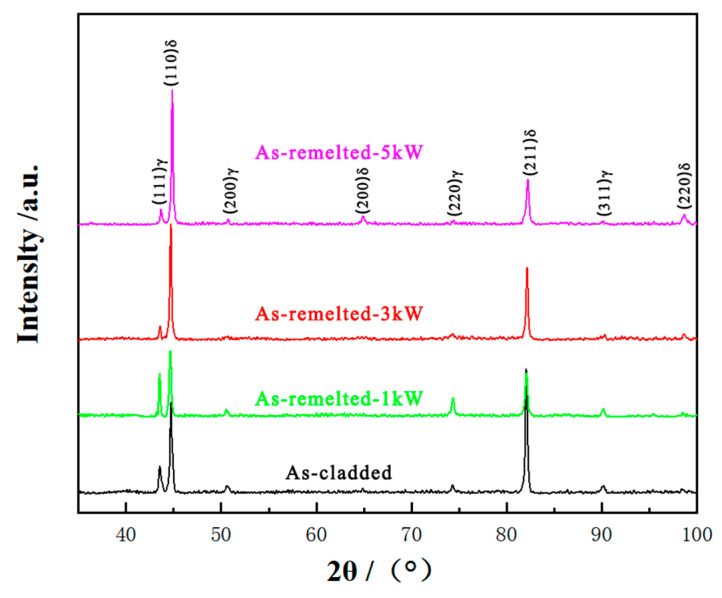
X-ray as-cladded and as-remelted coatings.

**Figure 6 materials-14-04965-f006:**
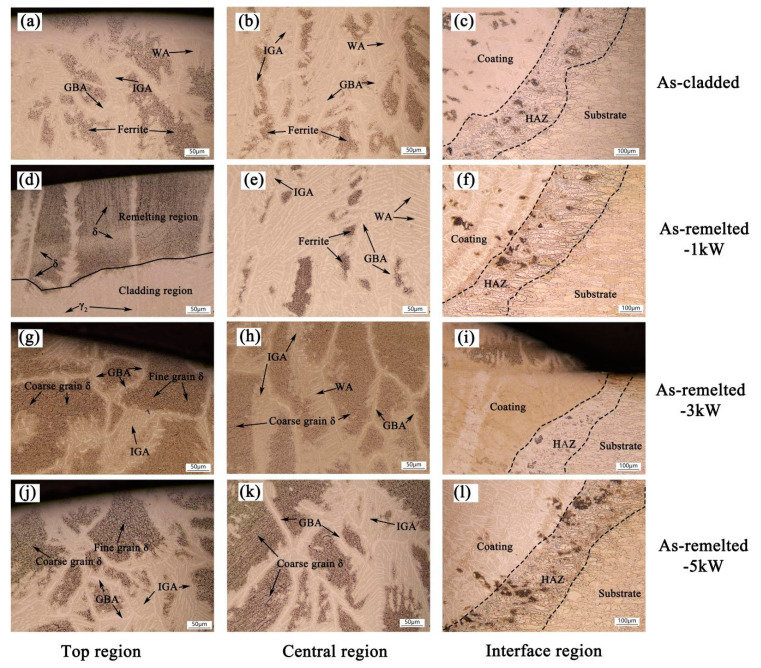
OM images of as-cladded and as-remelted coatings: (**a**) at the top region, (**b**) at the central region and (**c**) at the interface region of as-cladding DSS coating; (**d**) at the top region, (**e**) at the central region and (**f**) at the interface region of as-remelted-1 kW DSS coating; (**g**) at the top region, (**h**) at the central region and (**i**) at the interface region of as-remelted-3 kW DSS coating; (**j**) at the top region, (**k**) at the central region and (**l**) at the interface region of as-remelted-5 kW DSS coating.

**Figure 7 materials-14-04965-f007:**
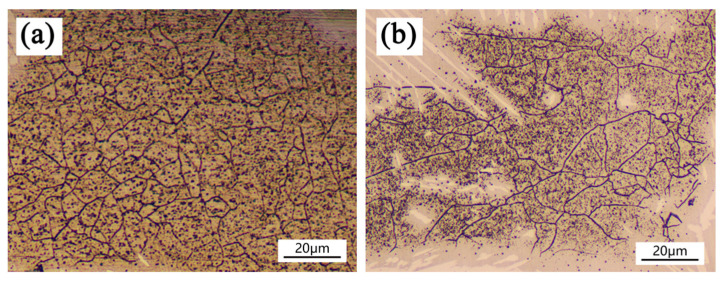
X-ray as-cladded and as-remelted coatings. (**a**): fine grain δ, (**b**): coarse grain δ.

**Figure 8 materials-14-04965-f008:**
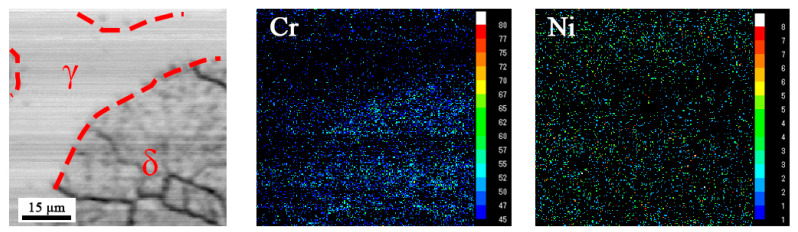
EPMA analysis of the as-remelted-3 kW sample.

**Figure 9 materials-14-04965-f009:**
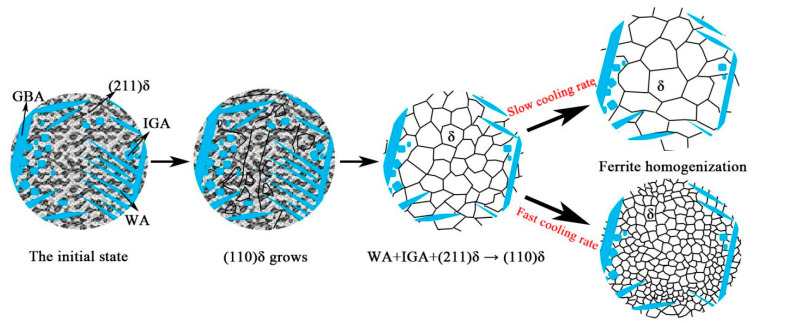
Diagram of the grain growth in as-remelted process.

**Figure 10 materials-14-04965-f010:**
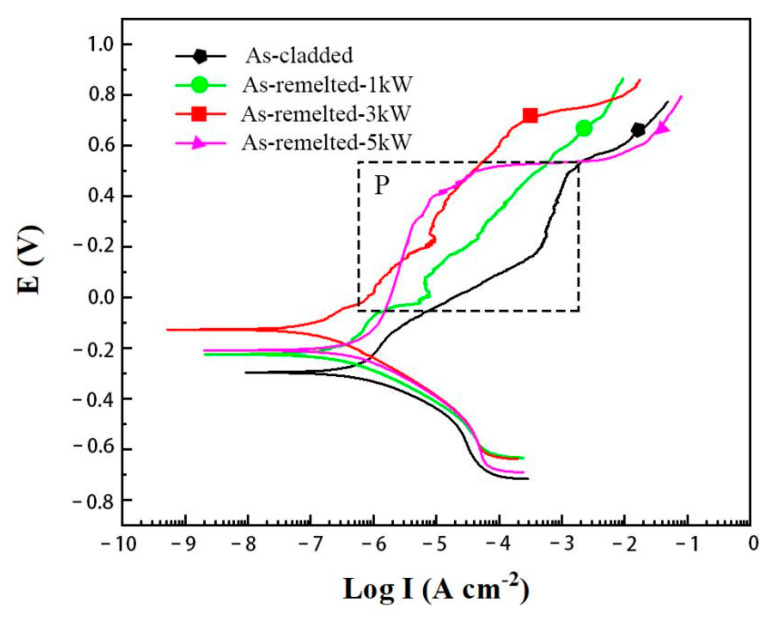
Polarization curves of as-cladded and as-remelted DSS coatings.

**Figure 11 materials-14-04965-f011:**
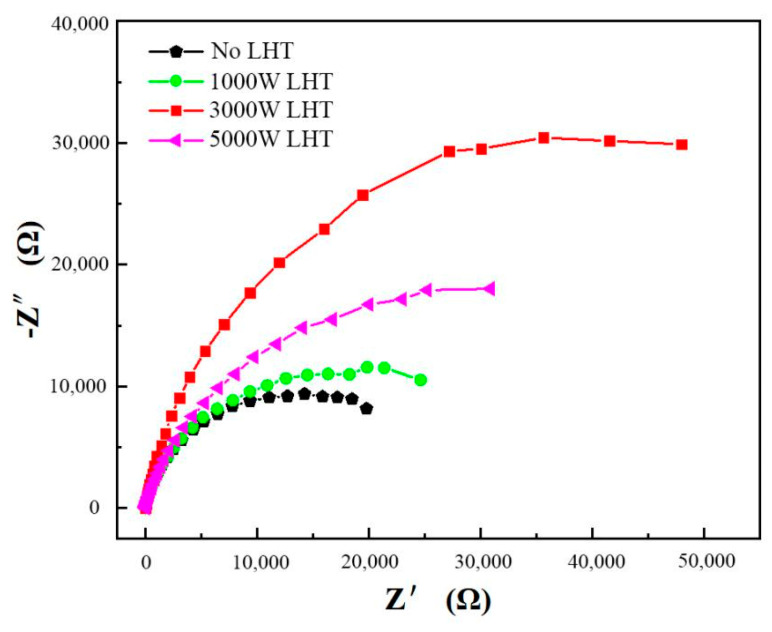
EIS curves of as-cladded and as-remelted DSS coatings.

**Figure 12 materials-14-04965-f012:**
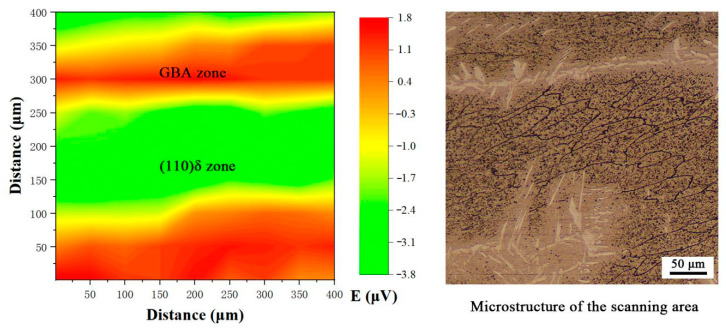
SVET result of the two-phase-coupled specimen in 3.5 wt.% NaCl solution.

**Table 1 materials-14-04965-t001:** Nominal chemical compositions (wt.%) of the S32101 duplex stainless steels.

C	Si	Mn	P	S	Cr	Mo	Ni	Cu	N	Fe
0.023	0.59	4.9	0.019	0.001	21.5	0.26	1.62	0.24	0.21	Bal.

**Table 2 materials-14-04965-t002:** Nominal chemical compositions (wt.%) of the ER2209 welding wire.

C	Si	Mn	P	S	Cr	Mo	Ni	Cu	N	Fe
0.012	0.35	1.59	0.015	0.001	22.56	3.05	8.62	0.06	0.15	Bal.

**Table 3 materials-14-04965-t003:** Remelting affected depth measurement results.

Sample	As-Cladded	As-Remelted-1 kW	As-Remelted-3 kW	As-Remelted-5 kW
Remelting Affected Depth (mm)	0	0.439	1.512	2.645

**Table 4 materials-14-04965-t004:** Polarization curve test results.

Sample	As-Cladded	As-Remelted-1 kW	As-Remelted-3 kW	As-Remelted-5 kW
Ecorr (mV)	−283.41	−222.86	−121.19	−196.36
Icorr (μA/cm^2^)	0.725	0.344	0.154	0.286

## Data Availability

Not applicable.

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
