# Peer review of "Microstructure and Corrosion Resistance of Underwater Laser Cladded Duplex Stainless Steel Coating after Underwater Laser Remelting Processing"

_materials, 2021, doi:10.3390/ma14174965_

Round 1
Reviewer 1 Report
In this work, Li et al. introduce a new technological process, based on a mixed underwater laser cladding and remelting, to improve the mechanical properties and the corrosion resistance of duplex stainless steel.
The aim of the work and the possible industrial applications of the implemented underwater process are both clearly described in the introductory section.
Morphological, structural and electrochemical investigations are rigorously performed, and methodologies used are carefully described. However, the presentation of the results, as well as the related discussion, should be improved before considering possible publication.
Authors should address some minor issues, as listed in the following.
Line 16: I would avoid to use the acronyms “WA”, “IGA”, “GBA”, as well as the symbol “δ” in the abstract, using instead the complete definitions “Widmanstätten austenite”, “intragranular austenite”, “grain boundary austenite” and “ferrite”.
Lines 39-40. For sake of clarity, as well as for a better introduction to their work, authors are invited to add here a few lines describing in brief the general approach used in underwater wet laser-based techniques.
Line 79. Define “δ” and “γ” symbols, and “DSS” acronym.
Section 2. When describing underwater processes ULC and ULR, more information should be provided. For instance, being the depth of water so crucial according to what authors state at line 95, at what depth the stainless steel samples were positioned in the pool for underwater treatments? What was the water starting temperature?
Figures 1 and 3. Define “BM”.
Line 159. Authors mention the “fish-scale” phenomenon, which should be briefly recalled here.
Line 155. Add “at different laser power values (1, 3 and 5 kW)” after “as-remelted coating”.
Line 163-165. Please check the statement “When the laser remelting parameter is 5000W, and the penetration depth of the remelting 165 affected zone (RAZ) exceeds that of the original coating.” It is truncated.
Line 164. For consistency with the rest of the paper, replace “5000 W” with “5 kW”. For the same reason, replace “3000 W” with “3 kW” at line 237.
Lines 183-185. Austenite peaks only increase at 1 kW, whereas they are lower in the as-cladded case and at 3 kW and 5 kW. Authors are invited to tentatively explain the reason of this peculiar behavior, namely that 1 kW seems to be a trade-off remelting power for favoring the formation of austenite phase. Also, for completeness, I invite the authors to discuss the significant increase of (110) ferrite phase with increasing remelting power.
Line 191. What do the authors mean with “The microstructure was analyzed according to reference [18]”? Please be more specific, describing in what terms the microstructure was analyzed.
Figure 6. Check the figure caption. It’s obviously left behind from the manuscript template.
Section 3.3. This section needs to be improved. First, a description of Fig. 8a should be given, indicating which is the austenite phase and which is the ferrite phase. Then, I cannot infer from EPMA maps a preferential concentration of Ni and Mn elements in the austenite phase. All maps seems to be spatially uniform, except from Cr map, for which the signal mostly concentrates in the grained (ferrite?) phase at the lower right sector. Authors are strongly invited to review this section.
Line 233. Please check the statement “By analyzing the microstructure and phase composition of the underwater laser remelting coating with different laser power densities.” It is truncated.
Line 237 and 248. For consistency with the rest of the paper, replace “3000 W” with “3 kW”.
Line 240. Please check the statement “The phase transformation of face-centered cubic austenite (fcc) to body-centered cubic (bcc) ferrite in duplex stainless steels during in-situ solution annealing heat treatment[18]”. It is truncated.
Figure 10. Replace “Volts” with “V” on the Y-axis. Replace “Log (I/Acm-2)” with “Log I (A cm-2)”. Define E and I in the caption.
Figure 11. Define Z’ and Z” as the real and imaginary parts of the measured impedance Z in the caption.
Author Response
Thanks a lot for your kind comments.Please see the attachment.

Reviewer 2 Report
Title: Microstructure and corrosion resistance of underwater laser cladded duplex stainless steel coating after underwater laser remelting processing
Honorable Authors,
Your paper presentes valuable investigations and interesting results. However, there are some misslacks, which have to be filled before publishing.
- From title and abstract there is not clear which of underwater welding method you used: wet welding, local cavity welding or dry welding. It has to be clarified.
- There is no description of underwater welding processes in the introduction. You have described only problems of investigated material. The introduction should present relevant scientific background to show necessity of your work and its relevance. You should show that in water environment the dry, local cavity, and wet welding methods are used. Morover, you should present topic to show, why you used one of these mehtods. Only in MDPI, there are some papers in relevant topic: https://www.mdpi.com/search?q=underwater+welding Moreover, the Special Issue is available https://www.mdpi.com/journal/materials/special_issues/underwater_processing_materials In my opinion, your article fulfills the aims of this SI very well.
- Have you tested the chemical composition of used materials or these values are nominal values? Mark in the table the source.
- Why these prameters were used (lines 115-117)?
- Why no standards' requirements were used in your investigations? Or you have not presented the relevant standard numbers in the text?
- The type of underwater welding method is not named in the text.
- Fig. 4 - I can see the photos from visual tests here.
- In many places, no spaces appeared before new sentence, e.g., "coating.During" - after dot, the space should be added. Please improve.
- No information about etching is presented near microscopic investigations.
- The sientific discussion is quite poor. Please compare your findings with papers published by other scientists more wider. What new hes been developed?
- You should support conclusions with quantitative results.
Author Response

(The authors gave the same response as above.)

Round 2
Reviewer 2 Report
Dear Authors,
The paper has been improved and it is sufficient for publishing.
I found two mistakes in names of Authors in references - ref. 2 and ref. 21.
In ref 2 should be:
Tomków, J. Janeczek, A., Rogalski, G., Wolski, A.
In ref 21 should be:
Kaierle, S., Overmeyer, L., Alfred, I., Rottwinkel, B., Hermsdorf, J., Wesling, V., Weidlich, V.
Author Response

(The authors gave the same response as above.)
